# Investigation of Photodynamic Therapy Promoted by Cherenkov Light Activated Photosensitizers—New Aspects and Revelations

**DOI:** 10.3390/pharmaceutics16040534

**Published:** 2024-04-13

**Authors:** Lisa Hübinger, Kerstin Wetzig, Roswitha Runge, Holger Hartmann, Falk Tillner, Katja Tietze, Marc Pretze, David Kästner, Robert Freudenberg, Claudia Brogsitter, Jörg Kotzerke

**Affiliations:** 1Department of Nuclear Medicine, University Hospital Carl Gustav Carus, Technische Universität Dresden, 01307 Dresden, Germany; 2Department of Radiation Therapy and Radiation Oncology, Faculty of Medicine and University Hospital Carl Gustav Carus, Technische Universität Dresden, 01307 Dresden, Germany; 3OncoRay—National Center for Radiation Research in Oncology, Faculty of Medicine and University Hospital Carl Gustav Carus, Technische Universität Dresden, Helmholtz-Zentrum Dresden—Rossendorf, 01307 Dresden, Germany; 4Helmholtz-Zentrum Dresden—Rossendorf, Institute of Radiooncology—OncoRay, 01328 Dresden, Germany

**Keywords:** photodynamic therapy, photosensitizer, Cherenkov light, rhenium-188

## Abstract

This work investigates the proposed enhanced efficacy of photodynamic therapy (PDT) by activating photosensitizers (PSs) with Cherenkov light (CL). The approaches of Yoon et al. to test the effect of CL with external radiation were taken up and refined. The results were used to transfer the applied scheme from external radiation therapy to radionuclide therapy in nuclear medicine. Here, the CL for the activation of the PSs (psoralen and trioxsalen) is generated by the ionizing radiation from rhenium-188 (a high-energy beta-emitter, Re-188). In vitro cell survival studies were performed on FaDu, B16 and 4T1 cells. A characterization of the PSs (absorbance measurement and gel electrophoresis) and the CL produced by Re-188 (luminescence measurement) was performed as well as a comparison of clonogenic assays with and without PSs. The methods of Yoon et al. were reproduced with a beam line at our facility to validate their results. In our studies with different concentrations of PS and considering the negative controls without PS, the statements of Yoon et al. regarding the positive effect of CL could not be confirmed. There are slight differences in survival fractions, but they are not significant when considering the differences in the controls. Gel electrophoresis showed a dominance of trioxsalen over psoralen in conclusion of single and double strand breaks in plasmid DNA, suggesting a superiority of trioxsalen as a PS (when irradiated with UVA). In addition, absorption measurements showed that these PSs do not need to be shielded from ambient light during the experiment. An observational test setup for a PDT nuclear medicine approach was found. The CL spectrum of Re-188 was measured. Fluctuating inconclusive results from clonogenic assays were found.

## 1. Introduction

Many different approaches can be considered to improve existing radiotherapy and radionuclide therapy [1,2,3,4,5]. The use of PSs, such as psoralens, as an approach to PDT is one of them. Psoralens and their derivatives (here: psoralen and trioxsalen) are able to intercalate between the base pairs of the DNA helix. After UV activation, covalent intra- and interstrand DNA crosslinks are induced, followed by the impairment of cell survival [6,7,8]. Although psoralen and trioxsalen are not chemotoxic (at the concentrations used in our experimental setup), upon activation, they form crosslinks and perform other binding processes with DNA bases, resulting in a phototoxic effect. General photoactivation with UVA is straightforward, but poses a problem when used therapeutically. Since most radionuclide therapies work deep within the patient’s tissue, external UV activation damages the interstitial tissue. This can be overcome by using CL for activation [9,10,11]. CL has a broad UV spectrum, with the highest intensity in UVA, and can be generated by diagnostic or therapeutic radionuclides when charged particles exceed the speed of light in a dielectric material.

Yoon et al. have proposed its use in radiotherapy and have generated CL using a clinical megavoltage radiation beam, which further activated trioxsalen [1]. They found large differences in survival between CL and non-CL setups. It has also been shown that CL is generated within the irradiated tissue by secondary electrons in the beam path [12], and CL is already being used for diagnostic purposes, mainly for visual imaging [13,14,15,16]. Additional experiments have shown that the light intensity increases with the photon energy dose and therefore high-energy radiation or radionuclides should be used [17].

In agreement with Yoon et al., we have recalled their experiments on clonogenic assays with an additional concentration point, trioxsalen AND psoralen as PSs and two other cell lines at our linear accelerator. The gel electrophoresis and absorbance measurements showed differences between the two PSs used. In accordance with these principles, it was interesting to try to transfer the approach of Yoon et al. to a nuclear medicine scheme, where instead of a radiation beam, a radionuclide is used for investigation. This is challenging because a radionuclide does not produce doses in the range of a megavoltage beam in the same time frame and therefore has a rather low dose (rate) setup. Re-188—a high energy beta emitter and standard radionuclide in nuclear medicine—is used for this work. It has a maximum beta energy of $E_{\beta ,max}=2.12$ MeV (but a low linear energy transfer of 0.19 keV/µm), resulting in a maximum tissue penetration depth of about 1.05 cm and a gamma radiation component at 155 keV that can be used for medical imaging. Advantageously, its half-life of about 17 h is sufficient for therapeutic use. Luminescence scans have acquired a spectrum of the generated CL, and in vitro studies were performed to determine the overall survival of the tumor cell lines FaDu, B16 and 4T1, representing hypopharyngeal tumor, melanoma and breast cancer, respectively. These experiments should be able to demonstrate a discernible effect of the PS-guided PDT approach.

## 2. Materials and Methods

### 2.1. Cell Lines

Several cell lines have been used for this setup.

FaDu cells (ATCC^®^ HTB-43^TM^) as a monolayer cell line were established in 1968 and correspond to the squamous cell carcinoma of the pharynx [18]. It has been used as a subcell line (FaDu_DD_) in radiobiological experiments since the 1980s [19]. The Department of Radiotherapy and Radiation Oncology, Medical Faculty, Technische Universität Dresden kindly provided the cells for our experiments. DMEM (Dulbecco’s minimum essential medium) containing 2% Hepes buffer, 1% of non-essential amino acids, 1% sodium pyruvate and 10% fetal calf serum was used for cell maintenance. 

B16-F10 (ATCC^®^ CRL-6475^TM^) is a cell line consisting of spindle-shaped and epithelial-like cells derived from mouse melanoma skin tissue. These cells have the potential for metastatic behavior and are highly motile in vitro [20]. They were cultured in DMEM–high glucose supplemented with 1% sodium pyruvate and 10% fetal calf serum.

The cell line 4T1 (ATCC^®^ CRL-2539^TM^) is a highly invasive mammary carcinoma cell line with characteristics similar to human breast cancer cells [21]. They were cultured in Gibco RPMI medium 1640 supplemented with 10% fetal calf serum.

For each cell line, the respective medium was changed every two days and used for cell splitting in preparation for experiments (in addition to trypsin and phosphate-buffered saline). Each cell line is proven to be free of mycoplasmas.

### 2.2. Photosensitizers

Psoralen (CAS: 66-97-7, purity ≥ 99%, Sigma Aldrich, Darmstadt, Germany) is the parent compound of the linear furanocoumarins. Its photosensitive properties are developed by the formation of monoadducts and covalent interstrand crosslinks with thymines (e.g., base of DNA) [22]. Trioxsalen (CAS: 3902-71-4, purity ≥ 99%, MedChemExpress, Monmouth Junction, NJ, USA) is a psoralen derivative with similar photoactive properties. However, these only occur when the PSs are activated by UV irradiation. As more UV light activates the PSs, more crosslinks are formed and reactions with other cellular structures and proteins are even possible [8,23]. The Lewis forms of both PSs are shown in Figure 1.

Psoralen and trioxsalen are already dissolved in dimethyl sulfoxide (DMSO). DMSO (CAS: 67-85-5, purity = 99.9%, Sigma Aldrich, Darmstadt, Germany) is a well-known chemical modulator and acts as a (here unwanted) radical scavenger. It is used at a concentration of 1% v/v as a solvent for psoralen. No chemotoxic influence of DMSO, psoralen and trioxsalen was found under our experimental conditions.

Concentrations of 10 µM, 50 µM and 100 µM of the respective photosensitizer were used. The absorbance spectrum of psoralen and trioxsalen was studied in relation to the influence of daylight using the TECAN Infinite^®^ M Nano absorbance plate reader (TECAN, Männedorf, Switzerland). The spectrum of psoralen in the wavelength range of 230 to 1000 nm was observed for a light exposure of up to 120 min. 

To determine if there were differences between the two photosensitizers, they were analyzed by gel electrophoresis. For this method, plasmid DNA was incubated with each PS and then irradiated with different doses of UVA light in an irradiation chamber (BS-02 Opsytec, Ettlingen, Germany) to activate the phototoxic behavior. The plasmid DNA was added to the gel with small pores and when the electric field was applied, the molecules or DNA fragments moved at different speeds depending on their size and charge. Thus, different formations (open circular, linear or supercoiled) of the DNA are found at different heights of the gel indicating the influence of the PS. The gel is inspected and optically evaluated with a charge-coupled device camera BioRad Gel Doc XR+ (BioRad, Feldkirchen, Germany).

### 2.3. Cherenkov Light and Irradiation

Irradiation experiments were performed using a 15 MV static photon beam (dose rate ≈ 6 Gy/min) from a clinical Versa HD^TM^ linear accelerator (Elekta AB, Stockholm, Sweden) at the Department of Radiation Therapy and Radiation Oncology, Medical Faculty and University Hospital Carl Gustav Carus, Technische Universität Dresden. For this, treatment planning was conducted using RayStation 10B treatment planning software (RaySearch Laboratories AB, Stockholm, Sweden) according to the clinical workflow based on a CT scan of the cell irradiation setup (see Section 2.6) and Monte Carlo dose calculation to irradiate the cells with a dose of 3 Gy and 6 Gy. A field size of 24 × 32 cm^2^ was chosen to achieve a homogeneous dose distribution for four well plates treated simultaneously.

For the radionuclide approach, a tungsten–rhenium generator (OncoBeta GmbH, Garching, Germany) was used to elute Re-188. This high-energy beta emitter is already in therapeutic use. It has a maximum beta energy of $E_{\beta ,max}=2.12$ MeV (mean energy 765 keV) and a gamma radiation component of 15.8% intensity, which is used for medical imaging. It has a half-life of 16.98 h and a mean linear energy transfer of 0.19 keV/µm. An activity of approximately 1.5 GBq Re-188 was used for the cell experiments. To estimate the corresponding dose, parameters such as the distance between the cells and the Re-188 culture flask, as well as the material between them, the gamma dose rate constant and the radioactivity dosage of Re-188 play a role. Calculations were performed similar to [26] and resulted in 13.9 mGy at the level of the cells for the given setup. When embedded in sufficient liquid medium, its decay radiation produces CL with a yield of 35 photons per decay event [27]. Light production was verified using the BioRad camera. The Thermo Scientific Varioskan LUX microplate reader (ThermoFisher Scientific, Darmstadt, Germany) additionally acquired a spectrum of the emitted CL. To block the CL, one side of the culture flask was masked with a black aluminum foil (Thorlabs BKF12, Bergkirchen, Germany; thickness: 50 µm). A planar image was acquired with an Anger gamma camera (Siemens Symbia Intevo T6) to ensure that the gamma radiation was not obstructed.

### 2.4. Colony Formation Assay

The survival fraction (SF) of the cell experiments is determined by a colony formation assay [28]. At the end of the exposure time, cells are removed from the irradiation setup, detached with trypsin, an aliquot is taken from each setup for the assay, and placed in an incubator to form cell colonies for six to seven days in a 24-well plate (beam) or 6-well plate (Re-188). The cells were then fixed in 4% formaldehyde and stained with crystal violet. Afterwards, the colonies were counted under a microscope at 25× magnification. Nine samples were analyzed for each setup with Re-188, and six samples were analyzed for the radiation experiments (like [1]). For all setups (non-irradiated, irradiated, irradiated but light blocked), colonies with more than 50 cells were scored as survivors, and the plating efficiency for each sample was estimated based on the initial number of seeded cells. Clonogenic cell survival was calculated as the relative plating efficiency of the treated versus the untreated samples.

### 2.5. Statistics

Each result is presented as the average of the samples in addition to the SD (standard deviation) based on each triplicate. Student’s *t*-test was used to ensure statistical significance. Two independent samples had a probability of error of p≤0.05. Statistical analysis was performed in OriginPro 2023b (64Bit SR1 10.0.5.157) and MS Office Excel 2019 MSO (16.0.10406.20006 32Bit). A Wilcoxon test was used to show the significance between the spectra measured with the TECAN reader.

### 2.6. Experimental Setup

To perform the radiation experiments, cells (600 or 1000 for 3 Gy or 6 Gy and controls, respectively) were plated in 24-well plates and transported to the facility in a Styrofoam box embedded with black aluminum foil to block temperature differences and ambient light after a preincubation with psoralen and trioxsalen for 1 h. They were irradiated with the beam coming from below. Between the cells and the beam, a culture flask of water was used as a “solid water” plate to generate CL according to Yoon et al. and the black aluminum foil was used to shield half of the well plates from CL for controls.

The following sketch shows the radionuclide setup used in the experiments (Figure 2). After the elution of Re-188, it is portioned (according to the specific activity) and injected into a T75 culture flask, adding distilled water to fill the flask completely. As a negative control without CL is required, one half of the culture flask is blocked so that no light can pass through. For the main experiments, another culture flask filled with plain water is placed on top of the Re-188 flask, blocking the beta particles to prevent any unwanted influence on the cells. The purpose of blocking beta particles is to ensure that there are as few other influences on the cells as possible other than gamma radiation (which cannot be blocked) and CL. The prepared cells are plated into 24-well plate which is placed on top of the water flask after a pre-incubation with the PS for 1 h. The 24-well plate provides the ability to test multiple setups in the same sample/experiment. The cells remain on the culture flasks for a 24 h exposure period in an incubator at 37 °C resulting in a dose of 25.1 ± 0.9 mGy (measured by optical stimulated luminescence (OSL) dose meters).

## 3. Results

### 3.1. Radiation Experiments

The FaDu, B16 and 4T1 cell lines were used to evaluate the clonogenic assays of Yoon et al. Concentrations of 50 µM and 100 µM psoralen or trioxsalen were used as PS. Figure 3 shows the survival fractions of six samples with and without a CL block as well as controls without PS. The mean survival fraction values are shown in Table A1 in Appendix A.

The differences between the cells with and without PS were analyzed for significance. The results are shown in Table A2 in Appendix A. The only significant difference between the transparent and blocked setup was for 50 µM trioxsalen at 6 Gy irradiation in the 4T1 cell line.

### 3.2. Photosensitizers Psoralen and Trioxsalen

Psoralen and trioxsalen (without photoactivation) are not chemotoxic at the concentrations used in these studies. However, to obtain reliable results, a negative control was observed for each sample and concentration. Survival fractions (SFs) were compared and there was no visible effect (p=0.33). Since DMSO acts as a radical scavenger and the trioxsalen used is dissolved in it, its properties and effects on the cells must be investigated beforehand. No relevant effect of DMSO was found (p=0.45). 

Considering that PSs are sensitive to light, it was investigated whether working in a laboratory with standard lighting affects or activates the properties of psoralen or trioxsalen and whether this presumed effect varies over time. An absorbance spectrum of the PS (concentration of 100 µM) was obtained using the TECAN reader (monochromator-based absorption measurement, wavelength range: 230 to 1000 nm, five light flashes with no rest time for 2 nm steps). The following figure (Figure 4) shows that there is no time/daylight induced effect on the absorbance/activation of psoralen (not significant by Wilcoxon test). Therefore, all experiments did not need to be performed in the dark.

In addition, gel electrophoresis was performed to determine whether both PSs have the same deleterious effects on plasmid DNA after exposure to UVA light (8 UVA lamps at 365 nm with an irradiance of 8 mW/cm^2^) in the irradiation chamber BS-02 for 5, 10, 20 and 30 min. The PSs react differently. Trioxsalen is more damaging than psoralen, resulting in more single- and double-strand breaks at the same concentration. Figure 5 shows an example of the proportions of supercoiled, open circular and linear plasmid DNA after 10 min of UVA irradiation (3.69 J/cm^2^). 

### 3.3. Cherenkov Light Production in Re-188

Re-188 produces CL in its own liquid environment (sodium chloride and distilled water) in the culture flask. One side of the culture flask is covered with a black aluminum foil and should not allow CL to pass through. The BioRad imager (ImageLab Software 6.0.1 b34) was used to verify this assumption (Figure 6). Analysis of the BioRad image shows that the black aluminum foil reduces the CL intensity to 7%, providing the desired light-blocking effect.

To verify that the gamma radiation is not affected by the black foil, a planar gamma camera image and a dose measurement using OSL dose meter [29] was performed. Figure 7 shows that the foil on the culture flask does not interfere with the homogeneous gamma radiation from Re-188 nor the incoming dose (p=0.350). There is no statistically significant difference in gamma radiation (p=0.351) between the two sides of the flask.

A spectrum of the CL produced by Re-188 was acquired using the Varioskan microplate reader. When compared to an empty well or a well filled with water, Re-188 produced a spectrum similar to that simulated for CL. The spectrum is shown in Figure 8.

### 3.4. Effects of Re-188 and Cherenkov Light

The effect of CL on the three cell lines FaDu, B16 and 4T1 was monitored for each experimental setup: irradiated samples without PS, and samples with different PS concentrations each a in transparent (CL visible) or covered (CL blocked) state. The following Figure 9 shows the survival fraction of the treated cells with 10 µM and 100 µM as the observed concentrations. The mean values of the samples are given in Table A3 in Appendix A.

The differences between with/without PS were analyzed for significance. No difference was significant. The results are shown in Table A4 in Appendix A. 

## 4. Discussion

Several studies and investigations have been performed using CL in clinical applications, not only for visual imaging [14,15,16] but also as a source of photoactivation for PSs such as psoralen [30]. Nevertheless, many conflicting results have been published [10,31,32,33,34,35] and several open questions remain.

Psoralen, as partially used in these investigations, is already used in the form of PUVA therapy (psoralen activated by UVA radiation therapy) and much research has been conducted [6,36,37,38,39,40,41]. Using a megavolt beam, UVA is induced by photon irradiation, which activates the phototoxic properties of psoralen [1,39,42,43,44,45]. Yoon et al. also used trioxsalen as a derivative of psoralen for their research in the field of PDT. They found that the SF was significantly lower when exposed to the CL, which activates psoralen-treated cells. 

However, this could not be confirmed in our studies. We investigated samples not only for trioxsalen but also for psoralen and added another concentration point. Yoon et al. showed the results for the B16 cell line. We performed the assays for the B16, 4T1 and FaDu cell lines, but did not see any relevant effects. On the contrary, the differences in SF for the negative controls were higher than the differences for the samples with PS, most likely due to a UVA-sensitivity. The SF of only one dose and concentration point for the 4T1 cells was significantly different between the transparent and blocked setups. This appears to be an outlier as all other results were not statistically significant. Nevertheless, this specific concentration and dose will be retained for further investigation.

When this approach was transferred to radionuclides in the field of nuclear medicine, no visible effect was found when different activation methods and the setups were investigated with radiation, light and radioactivity [39]. Although a similar setup was used, there are differences from the current investigations. In the previous studies, a small volume of only 20 µL (or 2 mL) was used to irradiate FaDu cells and produce CL. Here, a volume of approximately 350 mL (Re-188 in distilled water) and an additional water flask (another 350 mL) were used. Thus, a higher CL intensity is expected along with a higher applied dose (8 MBq [39] vs. 1.5 GBq) and longer irradiation times.

As previous studies have shown, “psoralen is not toxic and does not damage the DNA”, when not activated [39]. The Cherenkov radiation spectrum for an energy similar to the maximum beta energy of Re-188, the emission maximum of CL, is close to the absorption maximum of psoralen [46]. This suggests that psoralen activated by the CL produced by Re-188 should have a significant effect on the survival of the cells used in our experiments and should provide an opportunity to make this setup applicable to PDT in nuclear medicine. 

Experiments have shown that Re-188 produces CL. This was confirmed not only the BioRad imager but also by the Varioskan microplate reader. A clear spectrum of the emitted radiation/light from a Re-188 sample was measured and compared to simulated spectra found in the literature [46] and found to be as expected. 

The black aluminum foil used to block out CL was found to reliably block light. A residual of 7.3% of the emitted light can still be found on the “covered” side of the experimental setup. This is comparatively low, considering Yoon et al.’s obscuration of about 25%. However, light propagates isotopically and can be reflected off the walls of the culture flask walls, so a small amount of light can still reach the “dark” cells of the setup. The use of water flasks or solid water, as in the studies by Yoon et al., can be used not only to generate CL but also to block beta radiation to minimize the influencing components. It should be noted that CL is also produced in plastic material and water at energies below the CL threshold [47,48,49]. In a realistic scenario, e.g., in a tumor, the radioisotope and CL are much closer to the cells and therefore receive more dose and even more CL intensity. This cannot be reproduced with the chosen setup. 

A scintigram of the Re-188 filled culture flask showed a homogeneous distribution of the emitted gamma radiation and no significant difference between the transparent and the covered side of the flask. In addition, the dose reaching the cells is not influenced by the black aluminum foil. 

In further investigations of PSs, a measurement performed with the TECAN absorbance plate reader has shown that the PSs psoralen and trioxsalen do not need to be handled in the dark. Since PSs are photoreactive, it is natural to handle them in the dark so as not to distort their properties prior to the planned experiments. Absorbance measurements of psoralen samples that were exposed to bright daylight (approximately 1780 lux) for varying lengths of time showed no difference in activation for up to 2 h. The results showed no significant difference (p=0.47) between the absorbance of psoralen in the dark compared to psoralen exposed to daylight. This again shows that both PSs are only activated by UVA light and do not require dark laboratory conditions. This is in contrast to the “darkened” setups used by other research groups [50,51].

In addition, both PSs were analyzed by gel electrophoresis for their effect on plasmid DNA at different periods of UVA irradiation. The results showed that trioxsalen is more phototoxic than psoralen. At lower concentrations, trioxsalen already shows the same effects as psoralen at higher concentrations. In addition, if the concentration is high enough and the trioxsalen has been well activated by UVA, double-strand breaks are also visible, most likely caused by an excessive amount of single-strand breaks. This was not seen for psoralen at any concentration or exposure length. Similar conclusions for psoralen and trioxsalen were drawn by [52].

Although the Re-188 setup was tested in three cell lines with different concentrations of psoralen and trioxsalen, no consistent effects were observed. A positive trend for a phototoxic effect in the transparent setup compared to the covered setup is visible, but the results are not significant and negligible compared to the differences in the negative controls and the variations in the cell experiments. This was not expected since a reduction in survival was associated with the photodynamic effect of the PS. Apparently, CL alone has a phototoxic effect on cell survival. A possible sensitivity of the cells to UVA light could be a possible explanation [53,54].

Interestingly, each cell line reacted differently to the PSs. This was not expected. While the B16 cell line seems to show only slight effects for psoralen and not for trioxsalen (which is contrary to the results obtained by the gel electrophoresis), the FaDu cells show the best results for a high dose of trioxsalen. Differences in responses to derivative PSs between cell lines have not yet been observed yet. 

Yoon et al. suggested that their approach was “more efficient at inhibiting proliferation than inducing cell death” [1]. This could not be confirmed because the Re-188 setup differs in irradiation time and dose rates. 

In conclusion, the activation of psoralen and trioxsalen via the CL produced by Re-188 in the colony formation assay is unlikely because the expected effects were not observed. One difference in the experimental procedure between the radionuclide setup and the setup of Yoon et al. is the dose rate. The dose rate in the experiments of Yoon et al. is at least 1000 times higher (several Gy/min) than the dose rate of the Re-188 setup with the water flask (about 10 mGy/h). The longer exposure period could also mean that the repair mechanisms of the cells have time to repair some of the damage. The influence of the dose rate and time has also been shown in other publications [10,39,55,56], where it is stated that the higher the dose rate or photon yield, the higher the CL intensity. Glaser et al. however state “that it is unlikely that the emission of CL by radionuclides is a usable source for phototherapy” [27]. Since the same cell lines did not show a significant effect for the high dose rates in the radiation therapy setup, this could not be confirmed by our studies.

Although our results differ from those of Yoon et al., there are other studies claiming significant effects of CL using Yttrium-90 (high energy beta-emitter similar to Re-188 with an even higher photon yield), Fluorine-18 or other standard radionuclides [55,57,58,59,60]. 

Other recent approaches using CL to improve imaging during therapy [61,62] include detecting and noting changes in radionuclide probes in the patient’s body during surgery [63,64,65], imaging as a dosimetry validation [66] or even combining PSs with biovectors for targeted PDT [67,68]. PDT research is also moving into the field of combination with chemotherapy, noting that CL can activate chemotherapeutic agents [69,70].

Obviously, the research field of CL used for PDT is not exhausted, neither in terms of radiation nor in terms of radionuclides or PSs. 

## 5. Conclusions

In this study, a PDT approach using a linear accelerator (Yoon et al.) was further investigated and transferred to radionuclide therapy. The phototoxicity of psoralen and trioxsalen is activated by CL generated by the radionuclide Re-188. This approach is important to overcome the limitations of deep-tissue phototherapy in combination with nuclear medicine therapy.

The results of Yoon et al. could not be confirmed and further investigations with additional cell lines, a different PS and an additional concentration point did not show significant positive results/effects. A simple setup to transfer the approach of Yoon et al. to nuclear medicine was found. In vitro studies were performed confirming the radiotoxicity of Re-188, and the chemotoxicity of psoralen, trioxsalen and the radical scavenger DMSO was not observed. CL appears to affect the cell lines even in the absence of the PSs. However, no additional significant effect of PSs on the observed survival fractions was observed. A cell-line response to the CL-activated PSs could be expected, but appears to be cell-line dependent. 

Investigations with other PSs, or experiments including cell metabolism and luminescence measurements, may shed light on the open questions. In addition, PSs with a higher absorption in the CL spectrum [71] or nanoparticles that enhance the emitted light [45,72,73] seem promising to further explore this interesting topic and possibly improve the already existing PDT method.

## Figures and Tables

**Figure 1 pharmaceutics-16-00534-f001:**
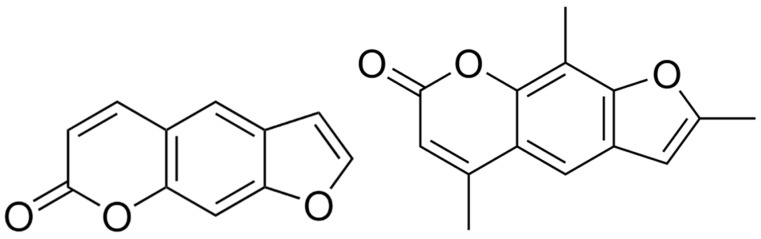
(**Left**): Lewis form of psoralen [24]. (**Right**): Lewis form of trioxsalen [25].

**Figure 2 pharmaceutics-16-00534-f002:**
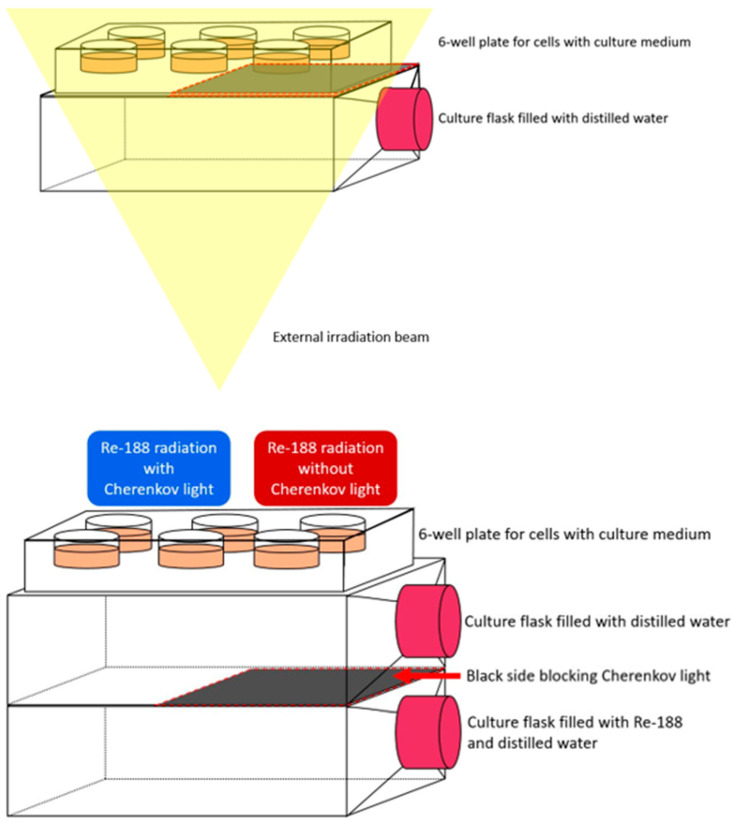
(**Top**): Setup when irradiating the cells at the linear accelerator. The beam irradiates the cells from below. The water flask in between is used as “solid water” to generate Cherenkov light. The black foil blocks the Cherenkov light on one side of the well plate. (**Bottom**): Schematic setup for the in vitro study of the effect of Cherenkov-activated PS on the different cells. At the bottom is the culture flask containing Re-188. On top is the culture flask containing water. At the top is a 6-well plate (for visualization, compare to the 24-well plate used in the experiments) for the cells. Each setup contains different concentrations of the photosensitizers and is determined in triplicate.

**Figure 3 pharmaceutics-16-00534-f003:**
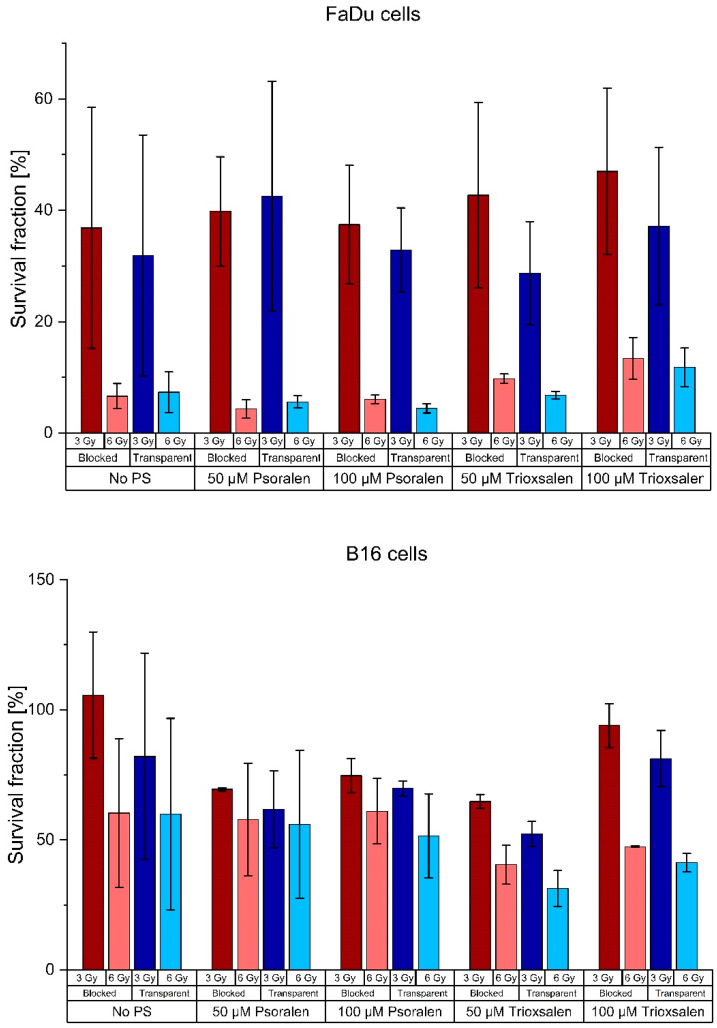
Survival fractions of the FaDu, B16 and 4T1 cell lines after irradiation with clinical 15 MV beam with and without CL blocked. The mean value ± the SD is shown for six samples.

**Figure 4 pharmaceutics-16-00534-f004:**
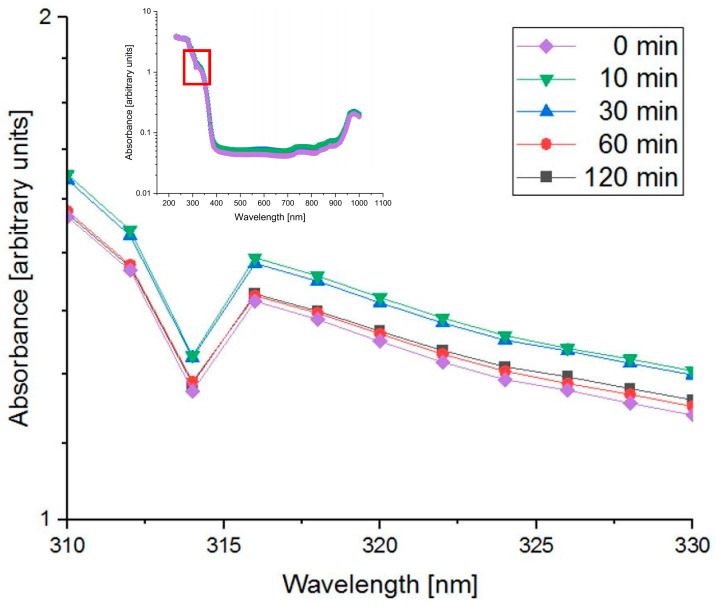
Absorption spectrum of psoralen (100 µM) as a function of exposure time to daylight (about 1780 lux). Although small differences are visible for 10 min and 30 min, these differences are not significant by the Wilcoxon test.

**Figure 5 pharmaceutics-16-00534-f005:**
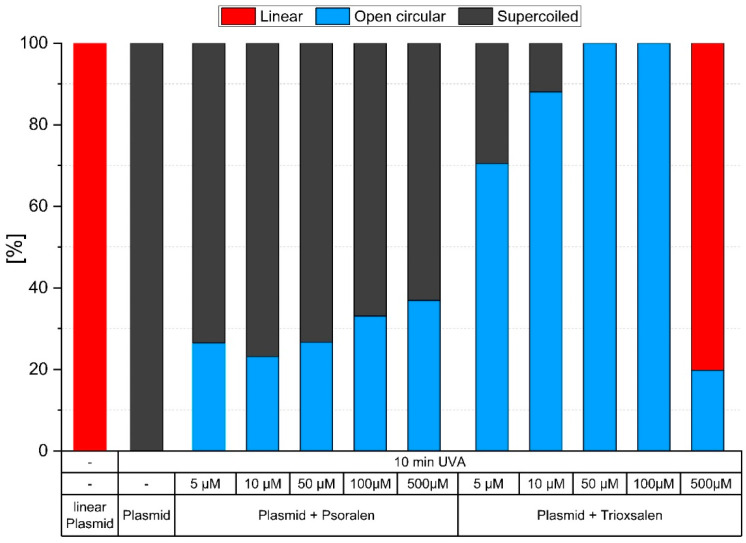
Gel electrophoresis analysis of proportions of supercoiled, open circular and linear plasmid DNA after 10 min of UVA irradiation with varying concentrations of psoralen and trioxsalen.

**Figure 6 pharmaceutics-16-00534-f006:**
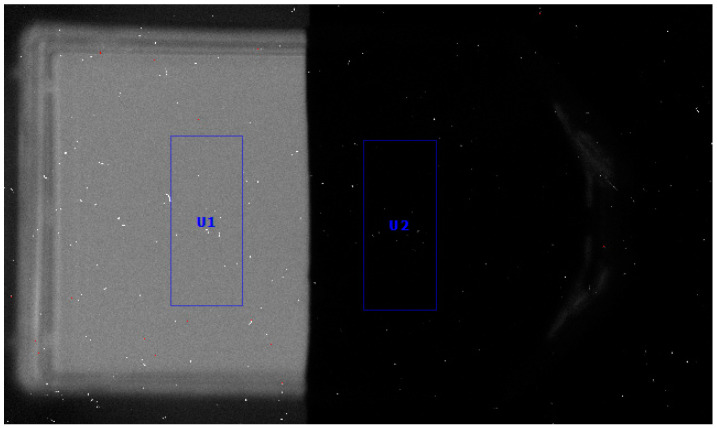
BioRad light intensity image of Cherenkov light generation in the culture flask and the culture flask covered with a black aluminum foil. Re-188 (2.1 GBq) was imaged for 300 s. Left side: Cherenkov light visible. Right side: Cherenkov light blocked by the foil. The light intensity is reduced to 7.3%. U1 and U2 are the two equal-sized reference regions of interest. The white spots are to be interpreted as direct hits by the BioRad camera.

**Figure 7 pharmaceutics-16-00534-f007:**
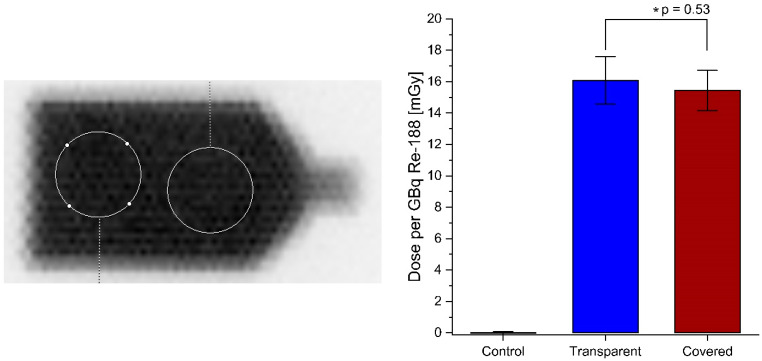
(**Left**): Planar scintigraphy of half-covered culture flask filled with 80 MBq Re-188 using a gamma camera (high energy collimator for $E_{\gamma ,Re-188}=155\;keV$). The image acquisition time was 5 min. Left side of the visible culture flask: covered with foil. Right side of the visible culture flask: transparent. There is no visible or statistically significant difference (p≈0.351) between the photon yields on the two sides. The circles indicate the regions of interest used to evaluate the potential differences of the two culture flask sides. (**Right**): OSL measurement of the dose reaching the cells. There is no difference between the transparent side and the covered side (p≈0.350).

**Figure 8 pharmaceutics-16-00534-f008:**
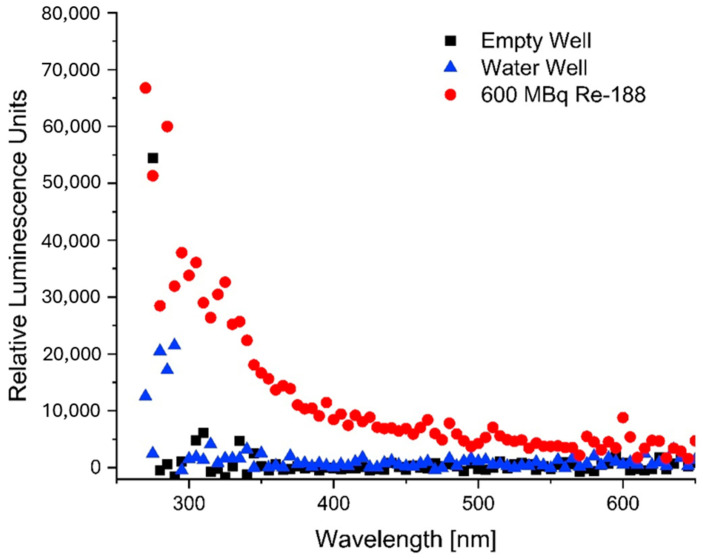
Spectrum acquired with the Varioskan showing the light emitted by an empty well (black), a well filled with water (blue) and a well filled with 600 MBq Re-188 (1 cm height in the well) for the wavelength range from 250 nm to 650 nm.

**Figure 9 pharmaceutics-16-00534-f009:**
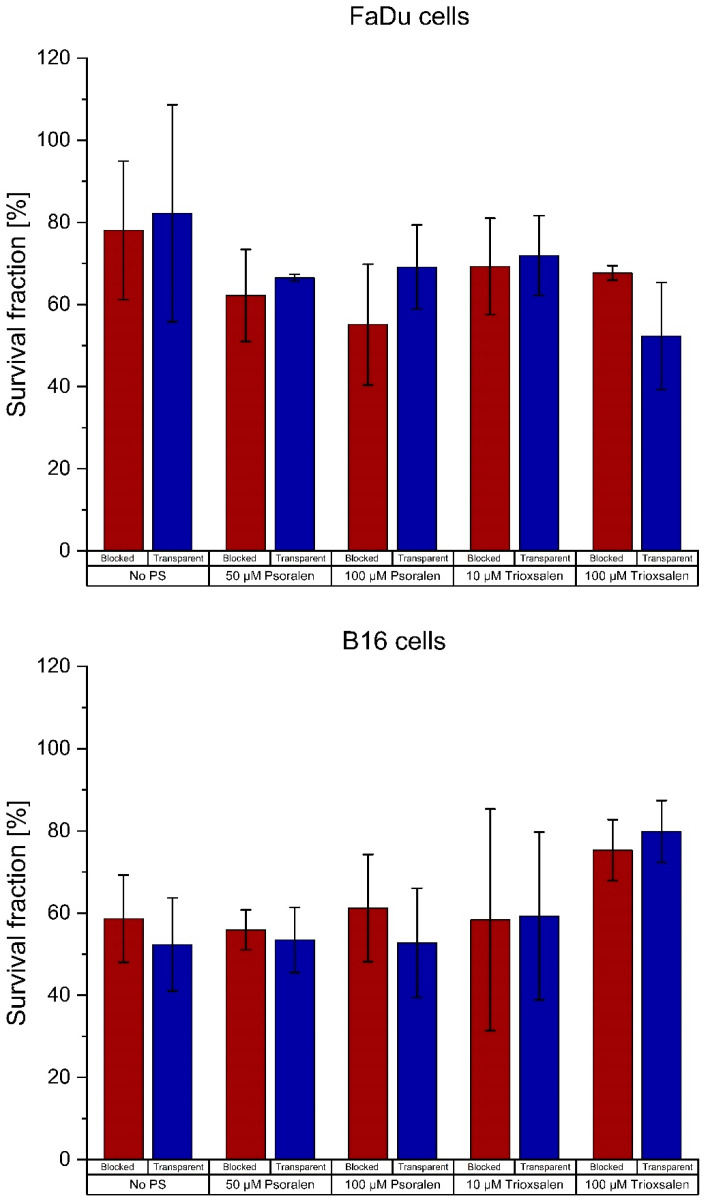
Survival fractions of FaDu, B16 and 4T1 cells. Each graph contains samples without photosensitizer and samples with the photosensitizer psoralen/trioxsalen for the transparent and covered setup. The mean value ± SD for nine samples is shown.

## Data Availability

The data presented in this study are available on request from the corresponding author.

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
