# Peer review of "Investigation of Photodynamic Therapy Promoted by Cherenkov Light Activated Photosensitizers—New Aspects and Revelations"

_pharmaceutics, 2024, doi:10.3390/pharmaceutics16040534_

Round 1

Reviewer 1 Report

Comments and Suggestions for Authors

This manuscript discusses the efficacy of photodynamic therapy (PDT) by activating photosensitizers (PSs) with Cherenkov light (CL) using external radiation and radionuclide therapy. This work intends to present new aspects regarding previous studies of the group. However, there are some issues that need to be addressed before publication. Below are some questions for the authors' consideration:

It is not necessary to define abbreviations throughout the manuscript. They must be defined only the first time they are cited. Please revise the manuscript and use abbreviations properly.

Please use time units throughout the manuscript: s, min, or h.

Lines 76-77: Please change the order of the cells. B16 is melanoma, and 4T1 are breast tumor cells.

Line 122: The authors report that the dose rate for external radiation was 6 Gy/min. This means that cells were exposed for 30 s and 60 s. Is this correct?

Line 154: Why were the cells seeded in different plates for external radiation and radionuclide treatment?

Line 170: Please add the number of cells plated.

Line 182: Please clarify why Beta particles were blocked.

Line 184: Change trioxsalen to PS since both psoralen and trioxsalen were tested.

Lines 186-187: Since cells were treated by Re-188 for 24 h, please add the final dose.

Figure 1: I did not understand why the authors used a 6-well plate in the figure if assays were performed with 24-well plates. In the legend, please change "FaDu cells" to "cells" since 3 different cells were used. I did not understand the phrase: "for visualization, compare to the 24-well plate used in the experiments".

Line 300: Please add information regarding the psoralen concentration and optical length to obtain the spectra.

Line 309: Please add the wavelength of UVA and the irradiance of the light source. Indeed, authors should add information in the methodology regarding this assay.

Line 361: Please define ROI.

Line 371: Please check the p-value in the figure.

Lines 517-518: I recommend rephrasing the sentence. Indeed, statistically significant differences were noticed for 4T1 cells using 50 microM of trioxsalen at 6 Gy. Authors should discuss this finding.

Line 602: Ref 34 does not support the use of Y-90 for Cherenkov radiation.

Reviewer 2 Report

Comments and Suggestions for Authors

Kotzerke and co-worker present in their submission to Pharmaceutics "Investigation of Photodynamic Therapy Promoted by Cherenkov Light activated Photosensitizers – New Aspects and Revelations". This is an interesting and carefully done study, which should be published, once the minor issues (see below) have been fixed.

The Lewis formula of psoralen and trioxsalen should be shown as an additional figure.

"The absorption range of psoralen/trioxsalen is between 320 nm and 400 nm. The absorption maximum is at 355 nm [24]." No absorption maximum of 355 nm is mentioned in ref. 24. The absorption maximum for psoralen is not at 355 nm, see 10.3390/molecules26092800, neither for trioxsalen (see Pharmazie (2005) 60, 3, 255).

The reference section is not consistently formatted. Sometime journal names are written in full length, sometimes they are abbreviated. The page numbers are also not consistently formatted, compare "3789-811" with "173-180".

Ref. 3, 68, 70, 72: the article numbers are missing.

Ref. 53: the year is wrong. Volume and page numbers are missing.

Ref. 67: volume and page numbers are missing.
